

# The application of two drainage angles in neurocritical care patients with complicated pneumonia: a randomized controlled trial

Anna Zhao[1], Huangrong Zeng[1], Hui Yin[1], Jinlin Wang[1], Wenming Yuan[1], Chao Li[1], Yan Zhong[1], Lanlan Ma[1], Chongmao Liao[1], Hong Zeng[1] and Yan Li[2]

[1] Department of Neurocritical Care Unit, Guangdong Sanjiu Brain Hospital, Guangzhou, Guangdong Province, China

[2] Department of Nursing, Guangdong Sanjiu Brain Hospital, Guangzhou, Guangdong Province, China

Corresponding author
Yan Li, liyan202305@126.com

## ABSTRACT

**Background.** Although head elevation is an early first-line treatment for elevated intracranial pressure (ICP), the use of the head-down or prone position in managing neurocritical patients is controversial because a change in a position directly affects the intracranial and cerebral perfusion pressure, which may cause secondary brain injury and affect patient outcomes. This study compared the effects of two postural drainage positions (30° head-up tilt and 0° head flat) on the prognosis of neurocritical care patients with complicated pneumonia and a clinical pulmonary infection score (CPIS) ≥5 points to provide a reference for selecting appropriate postural drainage positions for patients with pneumonia in neurocritical care units.

**Methods.** A prospective randomized controlled study was conducted with 62 neurocritical care patients with complicated pneumonia. The patients were categorized into control (=31) and experimental (=31) groups in a 1:1 ratio using a simple randomized non-homologous pairing method. Emphasis was placed on matching the baseline characteristics of the two groups, including patient age, sex, height, weight, Glasgow Coma Scale score, heart rate, mean arterial pressure, cough reflex, and mechanical ventilation usage to ensure comparability. Both groups received bundled care for artificial airway management. The control group maintained a standard postural drainage position of 0° head-flat, whereas the experimental group maintained a 30° head-up tilt. The efficacy of the nursing intervention was evaluated by comparing the CPIS and other therapeutic indicators between the two groups after postural drainage.

**Results.** After the intervention, the within-group comparison showed a significant decrease in the CPIS ($P < 0.001$); procalcitonin levels showed a significant decreasing trend ($P < 0.05$); the arterial oxygen pressure significantly increased ($P < 0.05$); the oxygenation index significantly increased ($P < 0.001$); and the aspiration risk score showed a significant decreasing trend ($P < 0.001$). A between-group comparison showed no significant differences in any of the indicators before and after the intervention ($P < 0.05$).

**Conclusion.** Postural drainage positions of 30° head-up tilt and 0° head-flat can improve the CPIS and oxygenation in patients without adverse effects. Therefore, we recommend that patients under neurological intensive care and having pneumonia be drained in a 30° head-up tilt position with good centralized care of the lung infection.

**Trial registration**. The study, "Study of Angles of Postural Drainage in Neurocritical Patients with Pneumonia," was registered in the Protocol Registration Data Element Definitions for Interventional Study database (# ChiCTR2100042155); date of registration: 2021-01-14.

## INTRODUCTION

In intensive care units, 60% of the patients have infections, of whom 64% have pulmonary infections (*Vincent et al., 2009*); these infections have 13% overall mortality (*Ferrer & Torres, 2018*). The primary nursing care management for preventing and treating pulmonary infections includes elevating the head of the bed to prevent aspiration, chest percussions for mucus clearance, postural drainage, adjusting the cuff pressure, and ensuring frequent oral care (*Isac, Samson & John, 2021*). Chest percussion, postural drainage, and lung expansion techniques are physical therapies widely used for mucus clearance in critically ill patients with pulmonary infections, which can reduce the mortality rate among critically ill patients (*Pozuelo-Carrascosa et al., 2018*). Head elevation is an early first-line treatment for elevated ICP, and most critically ill patients are advised to elevate their bed head by 30°. The use of the head-down or prone position in managing neurocritical patients is controversial because a change in a position directly affects the intracranial and cerebral perfusion pressure, which may cause secondary brain injury and affect patient outcomes (*Deng et al., 2020*). Moreover, evidence suggests that positional drainage is not an independently superior method for physical airway clearance (*Burnham, Stanford & Stewart, 2021*). Maintaining stable intracranial and cerebral perfusion pressures is frequently difficult based on measures to prevent and treat lung infections in neurocritical care patients. This study adopted a randomized controlled trial with a matched-pair design to investigate the effect of 30° head-up tilt and body position drainage on the clinical pulmonary infection score (CPIS), blood gas indicators, physiological monitoring indicators, and other indicators in patients with neurocritical diseases. This study aimed to recommend the best head elevation angle for postural drainage in neurocritical care patients with complicated pneumonia and to provide theoretical and practical evidence for ensuring a favorable patient prognosis.

## MATERIALS & METHODS

### Study design

This experimental, parallel, randomized, controlled research was conducted in a tertiary-level brain specialist hospital neurocritical care unit (NCU).

## Settings

The study sample in this prospective randomized controlled trial comprised 62 patients with neurological critical illness concomitant with pneumonia who had been hospitalized between January 2021 and December 2021.

## Research participants

Paired patients admitted from January 2021 to December 2021 to our hospital's NCU with neurocritical illness, concomitant pneumonia, and artificial airways were included in this study. Participants were categorized into the experimental and control groups using a 1:1 matched-paired design (Fig. 1). The baseline characteristics of both groups were considered, including patient age, sex, height, weight, Glasgow Coma Scale (GCS) score, heart rate (HR), mean arterial pressure (MAP), cough reflex, and mechanical ventilation usage.

## Inclusion and exclusion criteria

The inclusion criteria were as follows: (1) neurocritical care patients with artificial airways and pneumonia, aged 18–65 years, both sexes; (2) fulfilling the diagnostic criteria for pneumonia (*Infection Group of the Chinese Medical Association, Respiratory Diseases Branch, 2018*), including infiltrates on chest radiograph or computed tomography and clinical and laboratory findings, with a CPIS (*Xu, Liu & Yu, 2015*) of ≥5 points (8); (3) hemodynamically stable patients with consistent intracranial pressure (ICP)-related indicators (owing to differences in the patients' baseline conditions, stability was evaluated comprehensively from the following three aspects: the bone window, unchanged pupil size, and stable ICP monitoring indicators), where a change in body and bed-head positions did not affect patient stability and stable physiological monitoring indicators, and with a difference of <5% in blood pressure between both arms; and (4) after a review by the hospital ethics committee, patients or their legal representatives agreed to participate in the trial and signed informed consent forms. The exclusion criteria included the following: (1) patients with unstable conditions and a high risk of intracranial hypertension and brain herniation; (2) patients with severe liver or kidney dysfunction or severe underlying lung disease, such as chronic obstructive pulmonary disease and lung cancer; (3) patients who could not be placed in a head-tilt or head-flat position owing to limitations of their disease condition; and (4) patients who were at high risk for aspiration or had undergone esophageal, gastric, or lung resection within the past 6 months.

## Research methods

### Research instruments

(1) A Spiegelberg ICP monitor was used. (2) A Mindray T8 electrocardiogram monitor was used to track the physiological monitoring indicators of HR, systolic blood pressure (SBP), diastolic blood pressure (DBP), MAP, and oxygen saturation ($SpO_2$). (3) Electric beds with protractors for precise head angle measurements. (4) A medical flutter mucous-clearing device (Model YS8001) for mechanically-assisted mucus clearance. (5) A soft pillow was used to fix the patient's position when they turned to their side to increase their comfort.

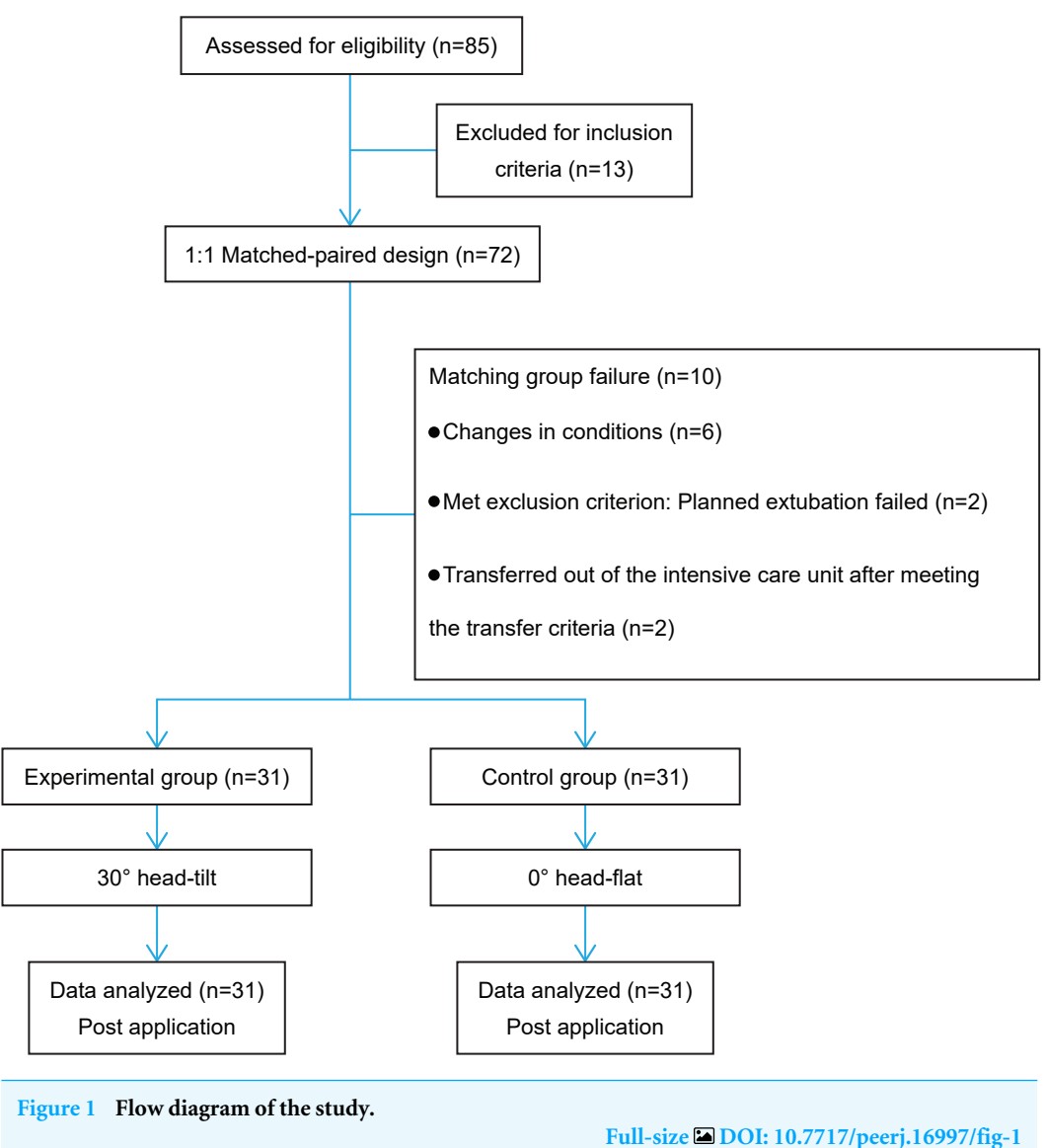

**Figure 1** **Flow diagram of the study.**

### Operating procedure

Patients in the experimental and control groups underwent mechanically assisted mucus clearance combined with percussion techniques using the following treatment process: (1) the identification of lung segments that required focused interventions based on imaging examinations and auscultation. When the patient was in the lateral position, an attempt was made to position the target lung segments in the upper position. (2) Patient preparation: patients receiving tube feeding were made to pause feeding for 30 min (*Bertoli et al., 2022*) in advance. For positioning, with the patient placed in the right lateral decubitus position, the procedure was performed by two operators. The operators stood on both sides of the patient, one near the head and chest and the other near the waist and buttocks. The operator on the left side hugged the patient's shoulders and waist with both hands, whereas the other one on the right side hugged the patient's waist and behind the knees

with both hands. Subsequently, with a numbered countdown from 1 to 3, they shifted the patient to the left side of the bed, bent the patient's knees, and turned them onto their right side. A soft pillow was placed in front of the chest, and another was placed between the legs for support. The patient's body was positioned laterally at an angle of approximately 100°. (3) An appropriate percussion head was chosen based on the patient's age and body surface area. (4) The treatment vibration frequency was adjusted to 25–30 Hz, and the vibration duration was set to 10 min. (5) Holding the handle of the percussion device with one hand, the percussion head was pressed firmly with the other hand to ensure that the percussion head was in close contact with the skin of the affected lung area on the chest wall (avoiding the spine, scapula, and kidney areas) before beginning vibrations. (6) Manual high-frequency percussion was performed on the back for 5 min (percussion frequency: 150–200 times/min; (*Xu, Chen & Wang, 2013*). (7) Suctioning: high-flow oxygen was administered for 30 s before suctioning (2 min for patients receiving mechanical ventilation with 100% oxygen concentration) using a closed suction system matched to the patient's endotracheal/tracheostomy tube size. The suction pressure was set to 20–26.7 kpa (*Shen & Xia, 2004*). Postural drainage was performed after mucus suction. The control group individuals were placed supine with the head of the bed at 0° for drainage (the bed head angle was changed from 30° to 0° at a rate of 5°/2 min to allow the patient to adapt to the position). The experimental group patients were positioned with the head of the bed tilted 30° for drainage. Both groups were maintained in their respective positions for 30 min. Notably, if the patient had a ventricular drainage tube, it was opened during postural drainage, and the preset height of the drainage tube was adjusted accordingly. In case the lateral position was related to the location of the patient's lung lesion and could not be uniformly auto-positioned, the blood pressure of the opposite upper limb was measured in the lateral position, and the opposite upper limb was placed on a soft pillow and elevated near the heart level.

## Evaluation indicators

SBP, DBP, MAP, HR, and SPO$_2$ were recorded after 30 min of postural drainage in the two positions. The mucus in the main airway was removed by suction after postural drainage in both positions. Blood gas analysis was performed 30 min later (*Yang & Xiao, 2019*). The pH, arterial oxygen pressure (PaO$_2$), and oxygenation indices were recorded. Laboratory indicators (C-reactive protein and white blood cell count (*Li et al., 2017*) were recorded before and after the intervention, and the CPIS was calculated (*Shen et al., 2019*). The total number of treatments received, the mean sputum volume drained, and the total number of days in the intensive care unit (ICU) were recorded for each of the two groups.

## Ethical considerations

The Ethics Committee of Guangdong Sanjiu Brain Hospital approved this study (IRB number: 2020-010-067, Approval date: July 10, 2020). Written permission (NO. A39202017) was obtained from the institution where the study was conducted, and written informed consent was obtained from all patients or their family members before the study.

**Table 1 Comparison of general and baseline data between the two groups $\bar{x} \pm s/\chi^2$.**

|  | Experimental group | Control group | T/$\chi^2$ | P |
|---|---|---|---|---|
| Age | 49.06 ± 10.41 | 48.45 ± 10.18 | 0.234 | 0.816 |
| Height | 169.52 ± 5.8 | 167.55 ± 5.57 | 1.363 | 0.178 |
| Weight | 68.56 ± 9.13 | 72.91 ± 10.91 | −1.701 | 0.094 |
| GCS | 5 ± 1.88 | 5.84 ± 1.98 | −1.708 | 0.093 |
| HR | 88.94 ± 20.44 | 85.94 ± 17.49 | 0.621 | 0.537 |
| MAP | 95.52 ± 14.25 | 96.84 ± 14.24 | −0.365 | 0.716 |
| Sex (male) | 28 (90.3%) | 28 (90.3%) | – | 1 |
| Coughing ability (+) | 30 (96.8%) | 31 (100%) | – | 1 |
| Mechanical ventilation | 27 (87.1%) | 30 (96.8%) | – | 0.354 |

**Notes.**

Abbreviations: GCS, Glasgow coma scale score; HR, heart rate; MAP, mean arterial pressure.

## Statistical methods

All data analyses were performed using IBM SPSS Statistics for Windows, version 26.0 (IBM Corp., Armonk, N.Y., USA). For the general information in the study, continuous and categorical variables were analyzed using independent-sample t-tests and chi-square tests, respectively. Within-group comparisons across the two groups before and after the intervention were performed using paired-sample t-tests or paired non-parametric tests. Between-group comparisons across the two groups were performed using independent sample t-tests or non-parametric tests. Furthermore, variations in physiological indicators in repeated measurements were analyzed using two-way repeated measures analysis of variance (ANOVA). Some patients reached the efficacy endpoint and were removed from the study early, whereas 48 patients were not removed early, and their measurements were taken 1–10 times before and after the intervention. Comparisons were made based on the number of pre- and post-intervention measurements. Statistical significance was set at $P < 0.05$.

## RESULTS

### General information of research participants

Overall, 72 critically ill patients with neurological disorders were included in this study. During the study period, five pairs of patients could not be matched because of changes in their conditions. Among them, three pairs withdrew because of changes in their conditions, one pair met the transfer criteria to be transferred out of the intensive care unit and could not continue the intervention, and one pair met the exclusion criteria because planned extubation failed on the third day of intervention for one of the paired patients. Finally, 62 patients completed the study (dropout rate, 13.8%). For the general information of the control and intervention groups, an independent sample $t$-test was used for continuous variables, such as age, height, weight, GCS, HR, and MAP, and a chi-square test was used for categorical variables, such as sex, cough reflex and use of mechanical ventilation. No significant differences were found between the two groups regarding age, sex, height, weight, GCS, HR, MAP, cough reflex, and mechanical ventilation usage ($P > 0.05$) (Table 1).

## Pairwise comparison of data

Independent samples $t$-test was used to compare the aspiration risk score, CPIS, white blood cell count, pH, PaO$_2$, partial pressure of carbon dioxide (PaCO$_2$), and oxygenation index of the experimental and control groups. The Mann–Whitney $U$-test was used to compare the procalcitonin (PCT) and C-reactive protein, and it was found that before and after the intervention, the levels of each of the above indexes did not differ significantly between the experimental and control groups ($P > 0.05$), with good comparability. A paired-sample $t$-test was used for the within-group comparison before and after the intervention across measures, such as the aspiration risk score, CPIS, white blood cell count, pH, PaO$_2$, PaCO$_2$, and oxygenation index. The risk of aspiration and CPIS in both the experimental and control groups showed a significant decreasing trend after the intervention ($P < 0.001$). The PaO$_2$ showed a significant increase ($P < 0.05$), the PaCo$_2$ demonstrated a decreasing trend ($P < 0.05$), and the oxygenation index significantly increased ($P < 0.001$) in both groups. Changes in the white blood cell count, C-reactive protein level, and pH were insignificant ($P > 0.05$). The Wilcoxon test was used for the within-group comparison of the PCT and C-reactive protein before and after the intervention. The PCT in both the experimental and control groups demonstrated a decreasing trend ($P < 0.05$). The changes in c-reactive protein demonstrated no significant differences between the two groups ($P > 0.05$) (Table 2).

## Changes in physiological monitoring indicators

Repeated measures ANOVA using the Bonferroni correction for multiple comparisons was employed to denote significant changes ($P < 0.005$).

The main effect of times of HR was not significant (F = 0.655, $P = 0.742 > 0.005$), implying that the difference in HR change was not significant in all patients at times 1–10; the main effect of pre- and post-intervention was not significant (F = 7.521, $P = 0.0092 > 0.005$), indicating that the difference in HR before and after the intervention was not significant in all patients. Additionally, the interaction effects of times with subgroups, pre- and post-intervention with subgroups, number of times *versus* pre- and post-intervention, and number of times *versus* intervention *versus* subgroups were insignificant ($P > 0.005$).

The main effect of times of SBP was not significant (F =0.905, $P = 0.5 > 0.005$), indicating that the difference in SBP change was not significant in all patients at times 1–10; the main effect of pre- and post-intervention was not significant (F = 0.643, $P = 0.427 > 0.005$), indicating that the difference in SBP between pre- and post-intervention was not significant in all patients; and the interaction effects of times and subgroups, pre- and post-intervention and subgroups, times and intervention and subgroups, the interaction effects of counts with pre- and post-intervention, and counts with intervention and subgroups were not significant ($P > 0.005$).

The main effect of times of DBP was not significant (F = 0.663, $P = 0.704 > 0.005$), implying that the difference in DBP change was not significant in all patients at times 1–10; the main effect of pre- and post-intervention was significant (F = 14.219, $P = 0.001 < 0.005$), indicating that the difference in DBP between all patients before

**Table 2  Pairwise comparison of the indicators.**

| | Control group | Experimental group | T | Z | P |
|---|---|---|---|---|---|
| Pre-aspiration risk score | 15.45 ± 2.93 | 14.87 ± 2.25 | 0.875 | | 0.385 |
| Post-aspiration risk score | 14.06 ± 2.29 | 13.52 ± 2.22 | 0.957 | | 0.343 |
| T | 5.079 | 5.044 | | | |
| P | <0.001 | <0.001 | | | |
| Pre-CPIS | 6.52 ± 1.12 | 6.45 ± 0.89 | 0.251 | | 0.803 |
| Post-CPIS | 3.9 ± 1.35 | 3.81 ± 0.83 | 0.34 | | 0.735 |
| T | 9.857 | 14.466 | | | |
| P | <0.001 | <0.001 | | | |
| Pre-white blood cell count | 13.55 ± 5.19 | 14.45 ± 8.42 | −0.507 | | 0.615 |
| Post-white blood cell count | 11.68 ± 4.26 | 11.81 ± 5.62 | −0.104 | | 0.917 |
| T | 1.786 | 1.698 | | | |
| P | 0.084 | 0.1 | | | |
| Pre-PCT | 0.27 (0.15,3.35) | 0.39 (0.25,3.90) | | −1.089 | 0.276 |
| Post-PCT | 0.17 (0.11,0.9) | 0.28 (0.16,0.73) | | −0.912 | 0.362 |
| Z | −3.072 | −3.472 | | | |
| P | 0.002 | 0.001 | | | |
| Pre-C-reactive protein | 3.37 (2.97,29.56) | 3.22 (2.93,29.7) | | −0.394 | 0.693 |
| Post-C-reactive protein | 3.25 (2.84,30.21) | 3.22 (2.07,29.88) | | −0.124 | 0.901 |
| Z | −1.321 | −1.184 | | | |
| P | 0.187 | 0.236 | | | |
| Pre-pH | 7.44 ± 0.06 | 7.42 ± 0.05 | 1.295 | | 0.2 |
| Post-pH | 7.43 ± 0.04 | 7.43 ± 0.04 | 0.763 | | 0.449 |
| T | 0.284 | −0.679 | | | |
| P | 0.778 | 0.502 | | | |
| Pre-$PaO_2$ | 106.75 ± 31.46 | 97.79 ± 28.28 | 1.18 | | 0.243 |
| Post-$PaO_2$ | 130.81 ± 40.07 | 114.85 ± 27.25 | 1.834 | | 0.072 |
| T | −2.605 | −2.563 | | | |
| P | 0.014 | 0.016 | | | |
| Pre-$PaCO_2$ | 36.11 ± 6.51 | 39.38 ± 7.06 | −1.896 | | 0.063 |
| Post-$PaCO_2$ | 43.15 ± 6.01 | 42.97 ± 6.67 | 0.113 | | 0.91 |
| T | −4.552 | −2.665 | | | |
| P | <0.001 | 0.012 | | | |
| Pre-oxygenation index | 223.44 ± 65.73 | 202.04 ± 51.93 | 1.422 | | 0.16 |
| Post-oxygenation index | 323.15 ± 83.03 | 305.59 ± 67.44 | 0.914 | | 0.364 |
| T | −5.411 | −6.82 | | | |
| P | <0.001 | <0.001 | | | |

**Notes.**

Abbreviations: PCT, procalcitonin; CPIS, clinical pulmonary infection score; $PaO_2$, arterial oxygen pressure; $PaCO_2$, partial pressure of carbon dioxide.

and after the intervention was significant. Moreover, the interaction effects of times with subgroups, pre- and post-intervention with subgroups, number of times *versus* pre- and post-intervention, and number of times *versus* intervention *versus* subgroups were insignificant ($P > 0.005$).

The main effect of times of MAP was not significant (F = 0.793, $P = 0.594 > 0.005$), implying that the difference in the change in MAP at times 1–10 was not significant in all patients; the main effect of pre- and post-intervention was not significant (F = 6.915, $P = 0.012 > 0.005$), implying that the difference in MAP before and after the intervention was not significant in all patients. Furthermore, the interaction effect of times with subgroups was not significant (F = 2.348, $P = 0.024 > 0.005$), and the interaction effects of pre- and post-intervention with subgroups, the number of times with pre- and post-intervention, and the number of times with intervention and subgroups were not significant ($P > 0.005$).

The main effect of times of $SPO_2$ was not significant (F = 1.448, $P = 0.209 > 0.005$), implying that the difference in $SPO_2$ change at times 1–10 was not significant in all patients; the main effect of pre- and post-intervention was not significant (F = 5.631, $P = 0.022 > 0.005$), implying that the difference in $SPO_2$ before and after the intervention was not significant in all patients. Moreover, in times *versus* subgroups, the interaction effects of pre- and post-intervention with subgroups, the number of times with pre- and post-intervention, and the number of times with intervention and subgroups were not significant ($P > 0.005$) (Table 3).

### Comparison of treatment and outcome indicators

An independent samples $t$-test was used to compare the treatment and outcome indicators between the two groups, showing that the total number of treatments, the mean sputum volume drainage, and the number of days in the ICU did not differ significantly between the intervention and control groups ($P > 0.05$) (Table 4).

### Adverse events

During the intervention process, no safety events, such as gastroesophageal reflux, aspiration, or hemodynamic fluctuations, occurred in either group.

## DISCUSSION

Using a 30° head-tilt position and a 0° head-flat position for postural drainage showed no differences in controlling pulmonary infections and caused no adverse effects to the patients.

Mechanically-assisted mucus clearance combined with manual percussion is a physical therapy primarily used for airway clearance in patients with pulmonary infections. The drainage position was selected based on the location of the pulmonary infection in the patient, with the affected area positioned upward and the drainage bronchial opening downwards to ensure effective mucus discharge, improve oxygenation, and promote recovery from pulmonary infections. This study used a matched-pair design with the CPIS of critically ill neurological care patients as the main outcome. After mechanically-assisted mucus clearance and manual percussion, the patients were categorized into the following two groups: 30° head-up tilt and 0° head-flat. The results showed that both positions could improve patient oxygenation, promote pneumonia recovery, and significantly reduce the CPIS within each group compared to before the intervention ($P < 0.001$). No significant

**Table 3  Analysis of repeated measurements of vital signs.**

| No. of times | Pre/post-intervention | HR | | SBP | | DBP | | MAP | | SPO$_2$ | |
|---|---|---|---|---|---|---|---|---|---|---|---|
| | | Experimental group | Control group | Experimental group | Control group | Experimental group | Control group | Experimental group | Control group | Experimental group | Control group |
| 1 | Pre | 87.95 ± 16.45 | 88.57 ± 17.35 | 133.14 ± 15.01 | 128.81 ± 20.95 | 81.09 ± 12.76 | 78.24 ± 11.93 | 97.09 ± 13.13 | 93.05 ± 13.15 | 99.32 ± 1.04 | 99.24 ± 1.37 |
| | Post | 89.27 ± 17.18 | 88.1 ± 20.5 | 134.59 ± 17.12 | 130.05 ± 21.38 | 78.86 ± 10.38 | 79.38 ± 12.97 | 96.45 ± 14.12 | 93.24 ± 13.99 | 99.73 ± 0.55 | 99.43 ± 0.87 |
| 2 | Pre | 86.86 ± 17.42 | 85.95 ± 16.55 | 134.95 ± 18.51 | 133.48 ± 14.55 | 80.5 ± 9.55 | 82.71 ± 10.12 | 98 ± 11.87 | 98.1 ± 10.99 | 99.5 ± 1.22 | 99.43 ± 0.98 |
| | Post | 87.59 ± 16.75 | 89.52 ± 16.65 | 133.45 ± 21.01 | 129.71 ± 15.25 | 78.95 ± 11.35 | 80 ± 11.34 | 94.05 ± 13.86 | 94.14 ± 11.28 | 99.55 ± 1.1 | 99.9 ± 0.44 |
| 3 | Pre | 89 ± 16.39 | 82.86 ± 13.42 | 138.55 ± 16.59 | 136.95 ± 14.98 | 82.5 ± 12.89 | 83.81 ± 12.15 | 98.68 ± 11.75 | 99.43 ± 12.83 | 99.55 ± 0.96 | 99.81 ± 0.51 |
| | Post | 90.95 ± 17.5 | 83.52 ± 14.6 | 137.23 ± 19.2 | 132.81 ± 18.77 | 81.32 ± 11.84 | 79.81 ± 13.96 | 98.59 ± 15.03 | 94.95 ± 13.68 | 99.73 ± 0.55 | 99.67 ± 0.8 |
| 4 | Pre | 90.14 ± 16.42 | 81.1 ± 13.95 | 129.41 ± 14.4 | 141.33 ± 16.7 | 78.77 ± 10.92 | 86.29 ± 11.28 | 94.59 ± 12.48 | 102.95 ± 13.45 | 99.73 ± 0.63 | 99.57 ± 0.75 |
| | Post | 90.91 ± 17.48 | 84.33 ± 14.63 | 135.41 ± 17.66 | 140.05 ± 21.07 | 77.55 ± 11.42 | 82.81 ± 12.24 | 95.77 ± 12.73 | 101.62 ± 16.33 | 99.86 ± 0.47 | 99.62 ± 1.07 |
| 5 | Pre | 87.32 ± 14.55 | 81.9 ± 17.26 | 132.77 ± 17.29 | 141.62 ± 12.57 | 80.73 ± 16.78 | 89.33 ± 15.4 | 95.77 ± 15.47 | 104.81 ± 14.01 | 99.64 ± 0.9 | 99.76 ± 0.62 |
| | Post | 89.05 ± 14.74 | 79.48 ± 17.77 | 133.68 ± 22.32 | 132.43 ± 16.61 | 77.14 ± 13.42 | 81.52 ± 10.25 | 92.86 ± 15.61 | 97.48 ± 11.91 | 99.82 ± 0.5 | 99.71 ± 0.78 |
| 6 | Pre | 90.14 ± 16.65 | 80.57 ± 16.77 | 126.91 ± 15 | 137.48 ± 15.15 | 76.18 ± 11.72 | 83.1 ± 13.45 | 92.23 ± 12.29 | 99.43 ± 14.9 | 99.86 ± 0.35 | 99.67 ± 0.8 |
| | Post | 91.77 ± 17.45 | 84.33 ± 17.89 | 136 ± 19.72 | 133.81 ± 20.95 | 80.14 ± 11.57 | 83.05 ± 15.95 | 98.73 ± 14.14 | 96.67 ± 18.31 | 99.64 ± 0.73 | 99.76 ± 0.54 |
| 7 | Pre | 89.91 ± 15.23 | 84.33 ± 18.24 | 133.68 ± 18.21 | 134.48 ± 15.87 | 80.5 ± 11.63 | 83.67 ± 12.23 | 94.68 ± 14.78 | 98.38 ± 14.06 | 99.77 ± 0.69 | 99.81 ± 0.51 |
| | Post | 89.77 ± 16.84 | 86.43 ± 19.35 | 130.73 ± 19.69 | 133.33 ± 19.44 | 75.64 ± 12.94 | 81.19 ± 13.52 | 92.59 ± 14.07 | 97.67 ± 15.4 | 99.73 ± 0.7 | 99.67 ± 0.73 |
| 8 | Pre | 87.32 ± 15.62 | 81.19 ± 17.3 | 128.14 ± 14.9 | 138 ± 14.89 | 79.64 ± 10.28 | 85.19 ± 8.99 | 94 ± 10.4 | 101.1 ± 10.01 | 99.55 ± 0.8 | 99.57 ± 0.93 |
| | Post | 88.73 ± 16.76 | 80.95 ± 17.65 | 130.5 ± 21.36 | 140.14 ± 17.55 | 76.23 ± 13.32 | 83.57 ± 11.97 | 91.86 ± 17.03 | 101.33 ± 14.09 | 99.45 ± 1.1 | 99.67 ± 0.66 |
| 9 | Pre | 88.68 ± 16.63 | 84.86 ± 19.02 | 131.45 ± 17.42 | 138.95 ± 12.52 | 81.23 ± 13.52 | 82.24 ± 10.21 | 100.95 ± 32 | 100.33 ± 9.9 | 99.68 ± 0.72 | 99.52 ± 1.08 |
| | Post | 90.45 ± 18.26 | 85.67 ± 20.57 | 131.5 ± 16.58 | 134.29 ± 16 | 80.05 ± 9.79 | 82.71 ± 15.74 | 97.18 ± 12.58 | 97.48 ± 14.43 | 99.91 ± 0.29 | 99.52 ± 0.93 |
| 10 | Pre | 87.36 ± 18.46 | 85.81 ± 16.99 | 130.18 ± 20.39 | 136.9 ± 14.22 | 77.95 ± 11.9 | 84 ± 10.47 | 94.14 ± 14.78 | 102.1 ± 12.51 | 99.64 ± 0.73 | 99.43 ± 0.98 |
| | Post | 89.5 ± 16.95 | 85.1 ± 18.47 | 127.55 ± 19.72 | 133.81 ± 16.23 | 72.86 ± 9.71 | 82.62 ± 12.06 | 88.68 ± 11.68 | 99.62 ± 13.12 | 99.77 ± 0.53 | 99.52 ± 0.93 |
| Count | F | 0.655 | | 0.905 | | 0.663/0.704 | | 0.793 | | 1.448 | |
| Count | P | 0.742 | | 0.500 | | 0.704 | | 0.594 | | 0.209 | |
| Count* Group | F | 0.947 | | 1.971 | | 1.715 | | 2.348 | | 0.696 | |
| Count* Group | P | 0.499 | | 0.062 | | 0.104 | | 0.024 | | 0.708 | |
| Pre/Post | F | 7.521 | | 0.643 | | 14.219 | | 6.915 | | 5.631 | |
| Pre/Post | P | 0.009 | | 0.427 | | 0.001 | | 0.012 | | 0.022 | |
| Pre/Post* Group | F | 0.124 | | 3.772 | | 0.019 | | 0.673 | | 0.178 | |
| Pre/Post* Group | P | 0.726 | | 0.059 | | 0.891 | | 0.417 | | 0.676 | |
| Count* Pre/Post | F | 0.48 | | 1.485 | | 1.634 | | 1.581 | | 1.599 | |
| Count* Pre/Post | P | 0.85 | | 0.172 | | 0.122 | | 0.162 | | 0.157 | |
| Count*Pre/ Post*Group | F | 0.907 | | 1.213 | | 0.885 | | 0.889 | | 1.677 | |
| Count*Pre/ Post*Group | P | 0.502 | | 0.295 | | 0.522 | | 0.545 | | 0.135 | |

**Notes.**
Abbreviations: HR, heart rate; MAP, mean arterial pressure; SBP, systolic blood pressure; DBP, diastolic blood pressure; SpO$_2$, oxygen saturation.

difference was found in the degree of improvement in the CPIS between the two groups ($P > 0.05$).

This study fully considered the rheological properties of mucus (*Zhang et al., 2018*). Mucus viscosity affects the difficulty and speed of clearance and drainage. Therefore,

**Table 4 Comparison of process indicators and length of ICU stay between the two groups ($\bar{x} \pm s$).**

|  | Experimental group | Control group | T | P |
|---|---|---|---|---|
| Total number of treatments received | $13.39 \pm 5.53$ | $13.55 \pm 6.19$ | $-0.108$ | 0.914 |
| Days in ICU | $5.4 \pm 1.56$ | $5.1 \pm 1.62$ | 0.757 | 0.452 |
| Mean sputum volume drained | $103.31 \pm 17.55$ | $100.77 \pm 14.22$ | 0.625 | 0.534 |

**Notes.**
ICU, intensive care unit

in this study, we ensured that the viscosity of the mucus was at level II and combined mechanically-assisted mucus clearance with manual percussion to ensure proper ciliary function. We believe this was also one of the main reasons why both the 30° head-tilt and 0° head-flat positions achieved satisfactory results in mucus clearance.

During the study, we strictly followed the guidelines for the nutritional management of critically ill patients (*Singer et al., 2019*) to control feeding-related risks. The strategy of mechanically-assisted mucus clearance and manual percussion involved stopping feeding 30 min before starting the procedure and allowing the stomach to empty. During the process, we strictly monitored the compatibility and pressure of the cuff with the airway (*Shu et al., 2022*), thereby avoiding adverse events, such as food reflux and aspiration.

The 30° head-tilt method was more suitable for managing critically ill neurological patients. The common positions for postural drainage include the semi-lateral position at 60°, head-down position, and prone position. The position was adjusted according to the actual lesion segment of the lung, following the principle that the affected lung should be placed in an upper position. However, focusing on mucus clearance, certain positions can increase the risk of food reflux and aspiration during postural drainage. This can be unfavorable for preventing ventilator-associated pneumonia (VAP) and managing pneumonia. Therefore, because of the presence of multiple drainage tubes in NCU patients and the influence of intracranial injury and underlying disease, it is necessary to elevate the head to 30 ° to maintain stable ICP and cerebral perfusion pressure. However, using the head-down position for postural drainage can increase the risk of ICP. Therefore, head-down or prone position postural drainage is relatively difficult to implement in critically ill patients and is considered a contraindication. Consequently, a personalized postural drainage plan should be developed based on the individual characteristics of critically ill patients under the premise of well-managed artificial airways. Furthermore, using a 30° head-tilt position for postural drainage can achieve the desired drainage effect, improve oxygenation, and be more conducive to controlling ICP and stabilizing cerebral perfusion pressure in NCU patients. This approach can reduce the risk of secondary cerebral perfusion injury caused by increased ICP due to body position changes.

This study had some limitations. First, there was no further classification of disease types because of the large number of types of neurocritical illness and the unequal probability of comorbid pulmonary infections. Second, the upper arm where blood pressure was measured was limited by the drainage position, resulting in the inability to measure blood pressure on the same upper arm. Therefore, there may have been some variation in the measurements. Third, due to the limitations of the financial support obtained for the

experiment and ethical considerations, the implantation of the probe and the measurement of the cranial pressure were not conducted for all patients during the experiment, and we measured and collected data on the cranial pressure only when the patient's treatment required the implantation of the probe; however, because the experimental data were collected from the monitoring of the cranial pressure in only three cases, they were insufficient to analyze the dynamic changes in the ICP. Therefore, we excluded this indicator from the analysis.

## CONCLUSIONS

Elevating the head of the bed by 30° and maintaining a position at 30° for 30 min after mechanically-assisted percussion and mucus suction can reduce CPIS and improve $PaO_2$ and the oxygenation index. In critically ill patients with pneumonia, a head-tilted position of 30 ° can reduce reflux and aspiration and decrease the incidence of VAP (*Mastrogianni et al., 2023*). Therefore, elevating the head of the bed by 30 ° for postural drainage after mechanically-assisted percussion and mucus suction is more consistent with the nursing requirements of critically ill patients and can promote recovery.

## ACKNOWLEDGEMENTS

We would like to thank all doctors and nurses of the Department of Neurocritical Care Unit, Guangdong Sanjiu Brain Hospital for their enthusiastic support and patient participation. We would like to thank Dr. Xie Zhiwei for his research support.

### Funding

The authors received no funding for this work.

### Competing Interests

The authors declare there are no competing interests.

### Author Contributions

- Anna Zhao conceived and designed the experiments, analyzed the data, prepared figures and/or tables, authored or reviewed drafts of the article, and approved the final draft.
- Huangrong Zeng analyzed the data, authored or reviewed drafts of the article, and approved the final draft.
- Hui Yin performed the experiments, prepared figures and/or tables, and approved the final draft.
- Jinlin Wang performed the experiments, prepared figures and/or tables, and approved the final draft.
- Wenming Yuan performed the experiments, prepared figures and/or tables, and approved the final draft.
- Chao Li analyzed the data, prepared figures and/or tables, and approved the final draft.

- Yan Zhong performed the experiments, prepared figures and/or tables, and approved the final draft.
- Lanlan Ma performed the experiments, prepared figures and/or tables, and approved the final draft.
- Chongmao Liao performed the experiments, prepared figures and/or tables, and approved the final draft.
- Hong Zeng performed the experiments, prepared figures and/or tables, and approved the final draft.
- Yan Li conceived and designed the experiments, authored or reviewed drafts of the article, and approved the final draft.

## Clinical Trial Ethics

The following information was supplied relating to ethical approvals (i.e., approving body and any reference numbers):

The Ethics Committee of Guangdong Sanjiu Brain Hospital approved this study (IRB number: 2020-010-067, Approval date: July 10, 2020).

## Data Availability

The raw data are available in the Supplemental File.

## Clinical Trial Registration

The following information was supplied regarding Clinical Trial registration:

ChiCTR2100042155.

## Supplemental Information

Supplemental information for this article can be found online at http://dx.doi.org/10.7717/peerj.16997#supplemental-information.

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
