# Peer review of "The application of two drainage angles in neurocritical care patients with complicated pneumonia: a randomized controlled trial"

_PeerJ, doi:10.7717/peerj.16997_

## Round 0.1 · original submission · Major Revisions

Dear Authors,

Please make corrections according to the reviewers' suggestions or write a detailed rebuttal on a point-by-point basis.

Reviewer 1 ·

Basic reporting

I have no objections to the mentioned references, structure, figures, or tables.

Comment on language and
grammar issues
I have no objections to language and grammar.

Experimental design

Some missing items in your results could affect the data analysis. In the results, I do not see the length of hospitalization of patients, which also affects complications. In your discussion, I suggest you provide the source or reason why the 0-degree head position takes 30 minutes.

Validity of the findings

I have questions for the authors:
1. Have you measured intracranial pressure?
2. How did you analyze the results related to PaO2 and O2 saturation values, which measurements, and in what time frame had an impact?

Additional comments

Although the results are convincing, the data analysis should be improved to give an answer about the superiority of the proposed conclusion.
The authors have chosen a topic that is undoubtedly of clinical importance.
The manuscript is clearly written. I see a weakness in the analysis of the missing data, which I mentioned so that the significance of the statistical data would be unequivocal.

Reviewer 2 ·

Basic reporting

I have no objections to the mentioned references. I have no objections to data and structure of manuscript.

Experimental design

1. Whether there was a statistically significant difference in the duration of hospitalization between the two groups?

Validity of the findings

Yours the results are convincing.

Additional comments

Interesting choice of research paper topic. The manuscript is clearly written. I see a minor weakness in data processing and selection of statistical analysis, but which do not affect the outcome of the research.

·

Basic reporting

Terminology in the paper should be consistent. For instance, "vital signs" were used in the Introduction and Method sections, but in the Results section, there's a different term, "physiological monitoring indicators." Similarly, seems the "adverse event" was referred to in some parts of the paper as "adverse results." It's important to use uniform terminology throughout the paper for clarity and precision.

Experimental design

1. Please specify the randomization method used to assign treatment (e.g. stratified randomization? Simple randomization?). Additionally, it is not clear how patients are paired during recruitment. Please clarify the pairing method/criteria in the abstract.
2. In Table 1, it is unclear to me which tests (t-tests or chi-squared tests) were performed to obtain each p-value. Please specify them in the table. Additionally, why are the test statistics not available for Gender, Mechanical Ventilation, and Primary Disease Condition?
3. In Table 2, please specify the specific test performed to obtain each p-value.
4. In Table 3, what is the meaning of “F/P”? What is the meaning of “text” and “contrast”?

Validity of the findings

1. Multiple statistical tests are performed on various evaluation indicators, which can increase the risk of false positive results when using unadjusted p-values. To mitigate this risk, please consider to apply the Bonferroni correction to control the family-wise error rate. Alternatively, authors may consider Benjamini-Hockberg procedure for controlling the false discovery rate.
2. In conclusion section, the sentence “Elevating the head of the bed by 30° and maintaining a position at 0° for 30 min after 269  mechanically-assisted percussion and mucus suction … “ appears to have a typo issue with “0°”.
3. In conclusion section, author mentioned “In critically ill patients with pneumonia, a head-tilted position of 30° can 271  reduce reflux and aspiration and decrease the incidence of VAP. “ Please provide the specific reference or study results that support this claim.

---

## Round 0.2 · Minor Revisions

Please make corrections according to comments from reviewer #3 or write a rebuttal.

Reviewer 1 ·

Basic reporting

No comment

Experimental design

No comment

Validity of the findings

No comment

Reviewer 2 ·

Basic reporting

No comment

Experimental design

No comment

Validity of the findings

No comment

·

Basic reporting

Thanks for the point to point response to the previous review question. The paper improved a lot after the revision.

I am still concerned about the p-values in Table 1, particularly for gender, coughing ability, and mechanical ventilation, without accompanying statistical analysis. Since these are key baseline characteristics, statistical testing is essential to demonstrate the balance between the two treatment groups. In addition, the inclusion of p-values suggests that some form of statistical testing has been conducted. Please explain how these p-values were derived if no statistical analysis was performed.

Experimental design

Looks good to me.

Validity of the findings

Looks good to me.

Additional comments

Looks good to me.

---

## Round 0.3 · accepted · Accept

Dear Authors,

The manuscript is now acceptable for publication in its current form.